# Analysis of Lambic Beer Volatiles during Aging Using Gas Chromatography–Mass Spectrometry (GCMS) and Gas Chromatography–Olfactometry (GCO)

**Katherine Witrick [1,2,](#)\***[🆔], **Eric R. Pitts [1]** and **Sean F. O'Keefe [2]**

[1] Department of Animal Science, Food & Nutrition, Southern Illinois University-Carbondale, Carbondale, IL 62901, USA; eric.pitts@siu.edu

[2] Department of Food Science and Technology, Virginia Polytechnic Institute and State University, Blacksburg, VA 24060, USA; okeefes@vt.edu

\* Correspondence: katherine.witrick@siu.edu; Tel.: +1-618-453-1766

**Abstract:** Lambic beer is produced using spontaneous fermentation. Gueuze is a style of lambic beer that blends "young" (1 year old) and "aged" (2+ years old) beers. Little is known about the development of volatile aroma compounds in lambic beer during aging. Solid-phase microextraction and gas chromatography–mass spectrometry were used to analyze volatile compounds from 3, 6, 9, 12, and 28-month-old commercial samples of lambic beer. Compounds were identified using standardized retention time and mass spectra of standards. Gas chromatography–olfactometry was used to characterize the aroma profiles of the samples. A total of 41 compounds were identified using gas chromatography–mass spectrometry (GC–MS). Ethyl lactate, ethyl acetate, 4-ethylphenol and 4-ethylguaiacol were identified in the 9, 12, and 28-month old samples. These four compounds have been linked to the microorganism *Brettanomyces*. Twenty-one aroma active compounds were identified using Gas chromatography–olfactometry (GC–O). As the age of the gueuze samples increased, a larger number of aroma compounds were identified by the panelists; the compounds identified increased from seven for the 3-month-old samples to nine for the 6-month-old samples, and eleven for both the nine and twelve-month-old samples, and seventeen for the twenty-eight-month-old samples.

**Keywords:** lambic beer; flavors; aging; gas chromatography–olfactometry

---

## 1. Introduction

Spontaneous fermentation has been used for the traditional manufacturing of beer, wine and apple cider. However, use of spontaneous fermentation is rare in beer manufacture because of the lack of control over the fermentation process; nevertheless, lambic beer is still being produced through spontaneous fermentation. Lambic beers are traditionally produced within fifteen kilometers of Brussels [1,2]. The Senne Valley and the surrounding region provide favorable conditions for spontaneous fermentation to occur. The concentration of fruit orchards and traditional farmhouse breweries provide an ideal environment for wild yeast and bacteria to become air borne so they can reside in the breweries that make lambic beer [3].

The brewing season for lambics begins in September and continues until sometime in April [4]. The tradition is to brew lambic beers during the colder months of the year to prevent the potential spoiling of the wort during the warmer summer months. The wort is allowed to cool overnight in open shallow trays (coolships) and is inoculated by the natural microflora present in the surrounding environment. Once the wort has finished cooling overnight, it is pumped into metal or wooden casks and stored in non-air-conditioned warehouses. The temperature in these warehouses can vary from 0 to 25 °C [1,2].

Lambic beers are unique in that they are still being produced using spontaneous fermentation. The fermentation process of lambics can be divided into four distinct stages, with microorganisms contributing key flavor and aroma compounds within each stage. *Hanseniaspora uvarum*, formally known as *Kloeckera apiculate* (a wild yeast) and enteric bacteria dominate the first stage of fermentation. The secondary stage of the fermentation process is dominated by *Sacchromyces sp.* followed by lactic acid bacteria (LAB) in the third stage. *Brettanomyces*, also known as brett, is a wild yeast often associated with the spoilage of red wines and ciders. In lambic beers, brett imparts expected, positive flavor characteristics and brett dominates the fourth and final stage of fermentation [5].

As noted earlier, the initial stage of fermentation for lambic beer is dominated by enteric bacteria and the yeast *H. uvarum.* The actual fermentation process begins three to seven days after the wort has been cooled. *H. uvarum* quickly reaches its maximum concentration of $10^5$ cells/mL within the first week of fermentation, but is quickly out-competed by *Saccharomyces*. *H. uvarum* only has the ability to ferment glucose and not maltose or more complex carbohydrates. *Enterobacteriacea* and *H. uvarum* are both fast growing microorganisms and they cause the pH of the wort to drop from around 5.1 to 4.6. This pH drop is due to the production of acetic and lactic acids [6].

The second and third stages of fermentation actually overlap one another. The third stage of fermentation begins three to four months after the beer has been brewed. The third stage is dominated by lactic and acetic acid bacteria both peaking around six to eight months, which usually correlates with the warmer summer months. Warmer temperatures are essential for the growth of both lactic and acetic bacteria [6].

The sourness associated with lambic beers can be contributed to by the presence of lactic acid produced by lactic-acid-producing bacteria [6,7]. The majority of the lactic acid present in lambic beer fall within the *Pediococcus* genus. While some strains of *Pediococcus* have been found to be beneficial in beer, others have the ability to produce a "ropy" surface layer that will leave the beer with a permanent haziness. While lactic acid is desirable, acetic acid bacteria as well as *H. uvarum* are undesirable in this particular style of beer because they each have the ability to convert ethanol into acetic acid, causing the beer to become acidic or hard. This only becomes an issue if the casks or barrels being used to store the beer are damaged or have leaks that allow oxygen to come in contact with the beer. The volatile acidity in commercial lambic beers ranges over 4-fold [6], which likely is related to differences in oxygen contact during storage [6,8].

The fourth and final stage is marked by an increase in the number of yeast cells. *Brettanomyces* plays an essential role in the development of the aromatic and flavor profile of lambic beer. Lambics, similar to other beers, are composed of a number of chemical and metabolic reactions [9]. A critical step in identifying what flavor and aroma compounds are present within a product is selecting the correct extraction technique. When selecting the best extraction technique, that technique must be capable of isolating all of the characteristic compounds that play a vital role in the overall flavor and aroma of the product without creating any artificial products [10]. Solid-phase microextraction (SPME) is one of the most commonly used extraction techniques for beer and other alcoholic beverages [11]. SPME has been found to be a suitable technique for the extraction of the volatile and semi-volatile compounds found in lambic beer when compared to other extraction techniques [12]. The majority of the volatile and semi-volatile compounds identified in lambic beers are a result of yeast metabolism [13].

The volatile and semi-volatile compounds found within lambic beer come from a number of different chemical groups, such as esters, acids, higher alcohols (fusel alcohols) and phenols, just to name a few [6,7]. Esters play an important role in the aromatic profile of this particular style of beer, which are by-products of a number of different microorganism [6,14]. Esters are well known for their fruity-floral aromas [13]. The two most influential esters produced in lambics are ethyl lactate and ethyl acetate, which originate from lactic and acetic acids during stage three [6,14]. Gueuze is result of the careful blending of different aged and different tasting gueuze beers [4,15]. Gueuze beers are a combination of young (1 year old) and old (2 and 3 years old) beer. Gueuze is similar to champagne in that they both undergo a secondary fermentation period in the bottle.

The purpose of this project was to characterize the volatile and semi-volatile compounds found in aging lambic beer samples through the use of headspace (HS–SPME) analysis using gas chromatography–olfactometry (GC–O) and gas chromatography–mass spectrometry (GC–MS).

## 2. Materials and Methods

### 2.1. Sample Acquisition and Storage

Different stages of fermenting gueuze beer were obtained from the brewmaster at Lindemans, a lambic brewery located in Vlezenbeek, near Brussels, Belgium. The samples collected were 3, 6, 9, 12, and 28 months old. The samples were taken from corresponding storage tanks within the brewery and placed into 120 mL sampling cups. Samples were stored at −40 °C prior to analysis.

### 2.2. Volatile Analysis

The extraction and analysis of the volatile and semi-volatile compounds in the lambic beer samples were conducted using solid-phase microextraction (SPME) following the method described by Thompson-Witrick et al., 2015 [12].

### 2.3. GC Parameters

GC was utilized to separate the volatile and semi-volatile. Compounds were separated using a nonpolar column (RTX-5MS, 30 m × 0.25 mm id.; 0.5 μm film; Restek, Bellefonte, PA, USA). Helium was used as the carrier gas at a flow rate of 1 mL/min. The GC oven had an initial temperature of 35 °C, which was held there for 5 min and then increased to 225 °C at a rate of 6 °C/min. Once the final temperature of 225 °C was reached, it was maintained for 10 min Thompson-Witrick et al., 2015 [12].

### 2.4. Identification of Volatiles

Linear Retention Index (LRI) values were used to help identify the volatile compounds present in the beer samples. A mixture of n-parafins (C5–C26) ASTM D2287 Quantitative Calibration Solution in carbon disulfide (Suplico, Bellefonte, PA, USA) was used in determining the LRI values for the volatile compounds eluded by the GC–O and the GC–MS. The databases Flavornet (http://www.flavornet.org/flavornet.html) and Pherobase (http://www.pherobase.com/) were used to aid in identifying the compounds based upon standardized retention and aroma.

GC–MS was used to aid in the identification of the volatile compounds. A Shimadzu GC-2010 Plus (Shimadzu, Columbia, MD, USA) with a GCMS-QP2010 Ultra mass selective detector and a SHRXI-5MS column composed of 5% diphenyl/95% dimethylpolysiloxane (30 m × 0.25 mm id × 0.25 μm film thickness) equipped with GCMSsolutions software and the NIST EIMS library were used.

### 2.5. Gas Chromatography–Olfactometry

2.5.1. Gas Chromatography–Olfactometry Training Parameters

An experienced 3-person (1 female and 2 males) sensory panel at Virginia Tech was used to evaluate the different stages of fermenting gueuze beer. The panelists were trained in eight, 15-min sessions before the study began. Thirteen pure aroma compounds associated with gueuze beer were selected and used to train the panelists. The training protocol was based off of Ferreira et al., 2003, procedure [16]. The aromas selected were: medicinal/barnyard, spice, sweaty, fruity, green, banana, brandy, banana-pineapple, citrus, vinegar, rancid, cheesy, and pineapple. Panelists were trained using pure compounds spiked in distilled water at 3, 1.5, 0.75, 0.5 and 0.25 (weak) times above the threshold. A lexicon was also developed during the training sessions. Once the panelists were comfortable with identifying and using the developed lexicon. They were asked to participate in the GC–O analysis. The panel was approved by the Virginia Tech Institutional Review Board (IRB #11-364).

### 2.5.2. GC–O Parameters

Five milliliters of beer were placed in a 15-mL glass vial fitted with a Teflon-lined cap. The sample was heated to 40 °C using an 'RTC basic' heater with an ETS D4 Fuzz Controller (IKA Werke, Wilmington, NC, USA) while being stirred using a 4-mm stir bar. A SPME fiber (50/30 μm Divinylbenzene/Carboxen/Polydimethylsiloxane (DVB/CAR/PDMS) on a 2-cm StableFlex fiber (Supelco Bellefonte, PA) was manually inserted into the vial and exposed for 30 min. A DB-5 capillary column (30 m × 0.25 mm id × 0.25 μm film thickness) was used for separating the volatile compounds. The same time and temperature methodology used for the GC–MS was also utilized for the GC–O. Those parameters are described above. The effluent coming from the GC column was split 1:1 ratio between the FID and the sniffing port (ODO-1 SGE, Nuremberg, Germany). Integration was done using a HP 3396A Integrator (Hewlett-Packard, Palo Alto, CA USA). GC–O runs were run as duplicate for each participant at each separate time point (3, 6, 9, 12, and 28 months).

### 2.5.3. Detection Frequency Method

A total of three panelists analyzed each sample twice, for a total of six values per sample. The detection frequency was summed based on off the odorant being detected at the same time by the panelists.

### 2.6. Statistical Analysis

Statistical analyses were conducted using coding for R (R Project for Statistical Programing, Vienna, Austria). Two-way multi variance ANOVA, coupled with a Tukey's HSD, was used to compare the compounds detected by a panelist vs. the different aged samples. Statistical calculations were carried out using a Kruskal–Wallis one-way analysis of variance test on total chemical groups compounds (esters, alcohols, acids, ketones, aldehydes, phenols) for the compounds detected by GC–MS.

## 3. Results

### 3.1. GC–O

For this study, the GC–O method detection frequency method was utilized. Three panelists carried out duplicate analysis for each aged lambic sample. For each sample, six individual analyses were obtained. The number of panelists who identified the odorant were counted and the odor descriptor was recorded. A total of 21 compounds were identified in all five samples. Compounds were identified based off of their calculated LRI values, mass spectra, and odor descriptors with those of pure compounds. Each compound identified in Table 1 Aroma compounds identified in aging lambic beer using GC–O has been previously reported in lambic beer [7,12]. A two-way multi variance ANOVA, paired with Tukey's HSD, was run to compare detected compound vs. aged sample, detected compound vs. detected compound, and age vs. age significant. Statistically, differences ($p = 0.005$) were seen between the compounds and the age. This was not surprising since there were differences in the number of times the compounds were detected by panelists in different samples. There were a number of instances when a compound would be identified in one sample and not in another. Statistically, differences ($p = 0.03$) were also observed between the different age samples. The could easily be explained by the fact that, as lambic beers begin to age, ester compounds usually associated with freshness will start to disappear, while aging compounds like ethyl lactate or aroma metabolites (4-ethylphenol) produced by *Brettanomcyes* during the fermentation process will be at a high enough concentration to be detected. Statistical differences ($p = 0.01$) were also observed between compound to compound.

**Table 1.** Aroma compounds identified in aging lambic beer using gas chromatography–olfactometry (GC–O).

| | | Age (Months) | | | | | | |
|---|---|---|---|---|---|---|---|---|
| LRI | Compound | 3 | 6 | 9 | 12 | 28 | Approximate Threshold ppm * | Descriptor |
| | | Frequency of Detection | | | | | | |
| 506 | Dimethyl sulfide | ND | ND | ND | 2 | 1 | 0.01–0.15 | sulfur, rotten |
| 593 | Diacetyl | ND | ND | 1 | 1 | 2 | 0.008–0.6 | buttery |
| 587 | Ethyl acetate | ND | ND | 2 | 2 | 1 | 0.5–50 | fruity |
| 658 | Propanoic acid | 3 | 4 | ND | 2 | 1 | 100–150 | rancid |
| 727 | 2-methyl-1-butanol (active amyl aclohol) | ND | 2 | ND | 4 | 2 | 50–65 | burnt |
| 745 | Isoamyl alcohol | 4 | ND | 2 | 2 | 2 | 40–70 | malt |
| 790 | Ethyl butyrate | 1 | 2 | ND | 2 | 2 | 0.05–0.3 | apple |
| 833 | Furfural (2-furanal) | ND | ND | 3 | 2 | 2 | 15–25 | bread |
| 875 | Isovaleric acid | 2 | 2 | 1 | 2 | 2 | 0.3–2.0 | rancid |
| 880 | Hexanol | 1 | ND | 1 | ND | 1 | 4 | green |
| 902 | Heptanal | ND | 2 | ND | 1 | 2 | 0.07–0.08 | rancid |
| 946 | Ethyl isohexanoate | 5 | ND | ND | 3 | 2 | ND | fruit |
| 1010 | Ethyl lactate | ND | ND | 1 | ND | 2 | 25 | fruit |
| 1019 | Hexanoic acid | 2 | 1 | 3 | 3 | 1 | 8–10.7 | sweaty |
| 1065 | Octanol | 2 | 2 | 1 | 1 | 1 | 0.9 | chemical |
| 1098 | Ethyl heptanoate | ND | 2 | ND | 2 | 2 | 0.7 | fruit |
| 1166 | Ethyl benzoate | ND | 2 | ND | 1 | 2 | ND | fruity |
| 1168 | 4-Ethylphenol | ND | ND | 3 | 2 | 3 | 0.3 | musky |
| 1204 | Decanal | ND | ND | 4 | ND | 1 | 0.006 | orange peel |
| 1241 | Ethyl phenacetate | ND | ND | ND | 1 | 1 | ND | fruit |
| 1253 | β-Phenethyl acetate | ND | ND | ND | ND | 1 | 2.5 | rose, sweet |

N = 6, * based upon literature values ASBC Beer Flavor Database, ND: no data, Frequency of detection.

When using a GC–O dilution method like CHARM (Combined Hedonic Aroma Response Measurement) and AEDA (Aroma Extraction Dilution Analysis) for the examination of key odorants, a series of dilutions are made usually between eight and twelve dilutions total. Multiple rounds of analysis (sniffing) are required for reliable detection. However, if the GC–O detection frequency method is utilized, a single analysis per sample is required without diluting the sample.

The primary flavor and aroma compounds of traditionally brewed beer come from the malt and hops. Lambics utilize several different strains of microbes to make its unique flavor and aroma profile. The aroma and flavor profile of spontaneously fermented or sour beers are defined by their acid profile. The sour aroma and tart flavor come from the acids, while the fruity aromas are esters that are produced by esterification between acids and alcohols. Acetic and lactic acid are the primary two acids that are detected when tasting this beer; however, the panelists were unable to detect acetic and lactic acid by GC–O. The DVB/CAR/PDMS SPME fiber has a useful polarity and absorption characteristics for characterizing and extracting a number of volatile and semi-volatile compounds; however, it does struggle with extracting some groups of compounds like acids.

Various yeast metabolites include aromatic and aliphatic alcohols, esters, organic acids, sulfur compounds, and a number of other compounds have been identified in beer. A number of these volatile and semi-volatile compounds have been shown to have a major impact on the flavor profile of a beverage even when presented in low quantities due to their low flavor threshold [17]. The most important flavor compounds in both lambic and other beers are the higher alcohols and esters, organic acids, dimethyl sulfide, and diacetyl; this is very similar to that of American lagers [18,19].

Among the twenty-one compounds detected, none were identified by everyone. Ethyl hexanoate was the only compound that had the highest detection frequency within the three-month sample. Only four (propanoic acid, 2-methyl-1-butanol, isoamyl alcohol, and decanal) odors were detected by 60% of the assessors.

### 3.1.1. Acids

Lambic beers are known for their high concentration of organic acids. Lambic beers are composed of not only lactic and acetic acid, but propionic, isobutyric and butyric acid, just to name a few. The panelists were only able to detect three different types of acids within the samples, propanoic, isovaleric, and hexanoic acid. Propanoic acid was found to play a larger role in the aroma profile of the beer in the initial stages of the aging process; however, as the beer is aged, it is picked up less often. Initially propanoic acid is detected by 50% of the panelists within the three-month-old sample and then increases to over 60% in the six-month-old sample, but decreases in the later samples. Although not identified as having as high of detection frequency as propanoic acid, isovaleric acid was detected in every sample. However, it was only detected by two panelists. Hexanoic acid, was the third and final acid identified by the panelists. Initially only two panelists were able to detect hexanoic acid in the 3-month-old sample, but, as the samples increased in age, the number of panelists able to detect it increased. However, only one panelist was able to detect hexanoic acid in the last sample.

### 3.1.2. Sulfur (Methyl Sulfunyl Methane)

DMS also known as dimethyl sulfide was only identified by the panelist in the twelve-month-old sample as well as the twenty-eight-month-old sample. For the twelve-month-old sample, only two panelists detected it, while only one person detected it in the twenty-eight-month-old sample. DMS in lambic and gueuze beer most likely comes from the metabolism of the Enterobacteria found in the wort [2]. The maximum concentration of DMS was reported by Van Oevelen and his colleges to be 450 ppb two weeks after the start of the main fermentation. The high concentration of DMS is quickly lowered by the stripping caused by the formation of $CO_2$ during the main fermentation. The average concentration of DMS normally found in bottles of gueuze is roughly 54 ppb (ranging from 25 to 75 ppb). There is no significant difference in the levels of DMS found in lambic beers and traditional lagers or ales [20].

### 3.1.3. Fusel Alcohols (Higher Alcohols)

Ethanol is just one of several alcohols that can be found in beers along with several other higher alcohols (fusel alcohols), which are considered important by products of fermentation [21]. Fusel alcohols also known as higher alcohols are some of the most abundant organoleptic compounds present in beers and lambics are no exception [22]. Four higher alcohols (isoamyl alcohol, 2-methyl-1-butanol (active amyl alcohol), hexanol, and octanol) were identified by the panelists. Octanol was one of the four higher alcohols detected by the panelists. In this study, only two panelists were able to detect octanol in the 3 and 6-month-old samples, while only one panelist was able to detect it for the remaining samples. Octanol has been previously been identified in several commercially available beers, the concentrations of which range from 0.03 to 0.97 mg/L. The threshold for octanol in beer, as reported by the American Society of Brewing Chemist (ASBC), is 0.9 mg/L [7,23]. The fact that octanol could be at or slightly below threshold could be one explanation of why so few of the panelists were able to pick up on the compound. Isoamyl alcohol and active amyl alcohol are often times represented as amyl alcohol. Amyl alcohol has been reported to be the most present and quantitatively significant higher alcohol found in beer [24]. Initially, isoamyl alcohol was detected by four panelists within the 3-month-old sample; however, as the samples increase in age, only two panelists were able to detect the presence of the compound. Active amyl alcohol, on the other hand, was only detected by two panelists in the beginning; however, as the sample increased in age, it was picked up by four panelists at twelve months, but, again, drops to two panelists at twenty-eight. Hexanol was the last higher alcohol identified by the panelists. Hexanol does not appear to play an important role in the aroma of lambic beers due to the lack of detection by the panelists.

### 3.1.4. 2–3 Butanedione (Diacetyl)

Diacetyl can be described as buttery, honey-like, and sweet. Diacetyl can be detected in small quantities in traditional lagers; however, strongly hopped or malted beers tend to mask the aroma [25]. The threshold for diacetyl in ales it is 0.1–0.4 ppm [18]. Diacetyl does not appear to play a crucial in the aroma of lambics in that it was only detected by one panelist in the nine and twelve-month samples and two for the twenty-eight-month sample.

### 3.1.5. Aldehydes

Decanal is initially not detected by the panelists until the 9-month-old samples, where it is considered to play an important role in the aroma of at 9-months due to it being detected by four panelists. However, seems to drop off after that, with only one panelist picking it up at twenty-eight months. However, heptanol, the other aldehyde detected by panelists, seems to not play such an important role in the aroma of lambics as was initially thought despite it being detected by two panelists at six months and then not again until twelve and twenty-eight months.

### 3.1.6. Esters

The concentration of higher alcohols in lambic beers and other styles of beer are similar; however, the concentration of esters is very different [1]. Of the twenty compounds detected by the panelists, seven were esters. They composed of the highest group of chemical compounds detected by the panelists. Despite ester's low concentration in beer, they do play a profound role in the overall flavor and aroma of beer and other fermented beverages [24]. However, of all of the ester compounds detected, none of them, with the exception of ethyl isovalerate, had more than three panelists able to detect the compound within any of the samples. Although ethyl acetate is found to be present in higher concentrations in lambic beers than any other styles [13]. The panelists were only able to detect it in samples 9, 12, and 28 months old. It should be noted that the GC–MS did detect ethyl acetate in all of the samples tested. The average concentration of ethyl acetate is between 8 and 48 ppm for traditional beers, 11.8 and 67.6 ppm for filtered gueuze and 60.9 and 167 ppm in unfiltered gueuze [7,13].

### 3.1.7. Phenols

4-ethylphenol, a compound produced by the wild yeast known as *Brettanomcyes* was the only phenol detected by the panelists. It was not detected by the panelists until the 9-month sample and beyond. *Brettanomyces,* is generally not detected in lambic beer until after about eight months, when *Brettanomcyes* is no longer competing with *Saccharomyces cerise.* Around nine months is when the concentration is high enough for *Brettanomcyes* to be detected, so it was not surprising that it was not detected by the panelists until 9-month point [5]. Thus, the overall concentration of 4-ehtylphenol would not be present or high enough to be detected by the panelists until then. Once, 4-ethylphonol was detected by 50% of the panelists [6]. Based upon previous sensory descriptions, 4-ehtylphenol does play an important role in the aroma of lambic beers and this data also suggest that as well.

### *3.2. SPME GC–MS*

SPME was used for the extraction of the volatile and semi-volatile compounds found in the aging gueuze samples. A total of 41 compounds were identified using a combination of retention index and mass spectral matching against library standards. A number of these compounds have been previously identified by Thompson-Witrick et al. [7,12]. GC–MS was utilized to characterize all of the volatile and semi-volatile compounds found within the aging samples. A Kruskal–Wallis one-way analysis of variance was run comparing the differences between the aging samples and each total chemical group. Statistical differences ($p < 0.05$) were observed between the aging samples of the different chemical groups with the exception of alcohols. It was not surprising to see statistically differences between the

total groups, because, as lambic beers age, the appearance and disappearance of compounds such as esters (ethyl lacate), acids (isovalerate), and phenols (4-ethylphenol and 4-ethylguaiacol) will occur [24].

Table 2, volatile and semi-volatile compounds identified in aging lambic beer using GC–MS, shows all of the compounds identified by GC–MS. A total of 40 compounds were identified. The volatile compounds identified came from a number of different chemical groups: esters (18), ketones (1), acids (5), alcohols (9), aldehydes (2), furans (2), monoterpene alcohol (1) and phenols (2). The number of compounds identified within each aged sampled differed slight 3 months (30 compounds), 6 months (27), 9 months (28), 12 months (28), and 28 months (36).

**Table 2.** Volatile and semi-volatile compounds identified in aging lambic beer using GC–MS.

| | | Age (Months) | | | | |
|---|---|---|---|---|---|---|
| | LRI | 3 | 6 | 9 | 12 | 28 |
| **Esters *** | | | | | | |
| Ethyl acetate | 587 | X | X | X | X | X |
| Ethyl lactate | 821 | X | ND | ND | X | X |
| Ethyl isovalerate | 862 | ND | X | X | X | X |
| Isoamyl acetate | 884 | X | X | X | X | X |
| Amyl acetate | 917 | ND | X | X | X | ND |
| Ethyl isohexanoate | 968 | X | ND | X | X | X |
| Ethyl hexanoate | 1000 | X | X | X | X | X |
| Isoamyl lactate | 1069 | X | X | X | X | X |
| Ethyl heptanoate | 1097 | X | ND | ND | ND | X |
| Ethyl octanoate | 1198 | X | X | X | X | X |
| Ethyl phenylethanoate | 1245 | X | X | X | X | X |
| β-Phenethyl acetate | 1255 | X | X | X | X | X |
| Ethyl nonanoate | 1296 | X | X | X | X | X |
| Ethyl decanoate | 1395 | X | X | X | X | X |
| Isoamyl Octanoate | 1456 | X | X | ND | ND | X |
| 2-methylbutyl octanoate | 1460 | X | ND | ND | ND | X |
| Ethyl undecanoate | 1494 | X | ND | X | X | X |
| **Ketones *** | | | | | | |
| 2-heptanone | 894 | ND | X | X | X | X |
| **Acids *** | | | | | | |
| Isovaleric acid | 861 | X | ND | ND | ND | ND |
| Hexanoic acid | 1000 | X | X | ND | ND | X |
| 2-ethylhexanoic acid | 1122 | ND | X | X | ND | X |
| Octanoic acid | 1180 | X | ND | ND | ND | X |
| Decanoic acid | 1366 | X | ND | ND | ND | X |
| **Alcohols** | | | | | | |
| Isopentyl alcohol | 700 | X | X | X | X | X |
| 2-methylbutanol | 744 | ND | X | X | X | X |
| 2-Heptanol | 905 | ND | X | X | X | X |
| Heptanol | 965 | ND | ND | ND | X | X |
| 2-Ethylhexanol | 1031 | X | ND | ND | ND | ND |
| 2-Nonanol | 1102 | ND | X | X | X | X |
| Phenylethyl alcohol | 1112 | X | X | X | X | X |
| Nonanol | 1172 | X | ND | ND | ND | ND |
| Decanol | 1272 | X | X | X | X | X |
| **Aldehydes *** | | | | | | |
| Nonanal | 1103 | X | ND | ND | X | ND |
| Decanal | 1198 | X | X | X | X | X |

| | | Age (Months) | | | | |
|---|---|---|---|---|---|---|
| | LRI | 3 | 6 | 9 | 12 | 28 |
| **Furans *** | | | | | | |
| 5,5-dimethyl-2(5H)-furanone | 954 | ND | ND | ND | ND | X |
| 2,5-Dimethyl-4-hydroxy-3(2H)-furanone | 1083 | X | ND | ND | ND | ND |
| **Monoterpene alcohol *** | | | | | | |
| Alpha.-Terpineol | 1190 | X | X | X | X | ND |
| **Phenols *** | | | | | | |
| 4-ethylphenol | 1166 | X | X | X | X | X |
| 4-ethylguaiacol | 1279 | X | X | X | X | X |

N = 6, X—denotes compound was identified, ND: no data, LRI: Liner Retention Index, * Indicates that statistical difference was seen within the chemical group ($p < 0.05$).

## 4. Discussion

*GC–MS*

Beer is a complex beverage, which is the result of a number of biological and chemical reactions resulting in the production of hundreds of flavor-active compounds [9]. However, a number of these compounds are produced during the fermentation process as metabolic intermediates or by-products of yeast [22]. Lambics are no different; however, a number of the acid compounds produced during the fermentation process are a result of lactic- and acetic-acid-producing bacteria [6]. Volatile and semi-volatile compounds identified in lambic beers come from a number of different chemical classes such alcohols, acids, esters, ketones, and phenols, just to name a few.

Thirty-one of the 41 compounds identified have not been previously reported in lambic beer until recently. Van Oevelen et al. and Spaepen et al. had previously identified a total of eleven compounds, such as isoamyl alcohol phenethyl alcohol, ethyl acetate, and ethyl lactate, caproic (hexanoic) acid, caprylic (octanoic) acid, capric (decanoic) acid, ethyl caprylate (octanoate), ethyl caprate (decanoate), and phenethyl acetate [1,26]. Witrick et al. have since further expand the number of compounds found within lambic beer [7,12].

Although the overall number of compounds identified by GC–MS remained relatively unchanged from sample to sample, the differences in relative concentrations did as well. Unlike a double IPA, lambic beers are allowed to continue to ferment as well as age for a minimum of a year and sometimes as long as three years [6]. Changes in the concentration of the volatile and semi-volatile compounds found within beer that occur over time are a result of the formation and degradation of chemical compounds. The formation of chemical compounds at concentrations above their flavor threshold will result in a change in the perceived flavor and/or aroma of beer, while the degradation of compounds may result in the loss of the initial freshness of the beer flavor [13]. It is well established that esters provide a fruity note to beer and are also considered an important an indicator of a beers age. Isoamyl acetate, an ester compound produced during the vigorous part of the fermentation process by yeast provides beer with a banana-like flavor. However, over time, particularly in storage, the concentration of isoamyl acetate and other esters can decrease below their threshold level, resulting in a loss of fruitiness within a beer [27]. On the other hand, there are certain ester compounds such as ethyl isovalerate (ethyl 3-methylbutyrate), ethyl isobutyrate (ethyl 2-methylpropionate), and ethyl lactate are synthesized during beer aging [28,29]. In this study, the development of ethyl isovalerate is not seen until month six; however, ethyl lactate is identified in all of the samples. The sudden appearance of 5,5-dimethyl-2(5H)-furanone in the 28-month-old sample is not surprising since previous research by Narziss et al. (1993) found that it only appears in beers that have undergone a prolong storage period or aging [30].

The concentrations of the volatile and semi-volatile compounds like organic acids, esters, fusel (higher) alcohols, aldehydes, and ketones, even diacetyl, will all contribute to the overall flavor and

aroma of the finished product [31]. The volatile compounds identified using GC–MS came from a number of different chemical groups: esters (18), ketones (1), acids (5), alcohols (10), aldehydes (2), furans (2), monoterpene alcohol (1) and phenols (2). The largest number of volatile and semi-volatile compounds identified came from the chemical group known as esters. Esters are considered trace compounds in comparison to ethanol and carbon dioxide. Esters are present in a relatively small concentration overall and tend to play an important role overall in the aroma profile of the finished product. The reason for this is due to esters' low odor threshold in beer [13]. Thus, it is crucial that brewers try to balance their ester concentration with other flavor compounds [22].

Esters are primarily formed during the vigorous phase of the fermentation process by the enzymatic condensation of organic acids and alcohols [13,32]. Esters specifically are synthesized within the intracellular space of the fermenting yeast cell [24]. Esters can be divided into two major groups: acetate esters and medium-chain fatty acid (MCFA) ethyl esters. Acetate esters are produced from acetic acid with ethanol or a higher alcohol [13,32]. The GC–MS analysis identified a total of six acetate esters; however, only ethyl acetate, isoamyl acetate, and phenyl ethyl acetate were identified in all of the aged samples. Amyl acetate was identified in samples 3–28 months, while 2- methylbutyl acetate was only identified in the 28-month-old sample.

Although a dozen or so esters have been identified in beer, only six of them are considered important in having a major contribution to the aromatic profile: ethyl acetate (solvent-like aroma), isoamyl acetate (banana), isobutyl acetate (fruity), phenyl ethyl acetate (rose and honey), ethyl hexanoate (sweet apple), and ethyl octanoate (sour apple) [13,32]. Lambics, unlike most other beers, can be aged for extended periods of time, allowing for a number of complex chemical reactions to continue [6]. Of the six major acetate esters, only ethyl acetate and phenyl ethyl acetate were identified by the panelists; however, isoamyl acetate, ethyl hexanoate, and ethyl octanoate were also identified by GC–MS. Ethyl acetate can be found in much higher concentrations in lambic beers than any other styles [13]. Ethyl acetate has a solvent-like fruity aroma with an odor detection threshold level of 30 ppm [13,33]. The average concentration of ethyl acetate is between 8 and 48 ppm for traditional beers, while it is 33.4–67.6 ppm for filtered gueuze and 60.9–167 ppm in unfiltered gueuze [13] Ethyl acetate was only detected by the panelists at 9 months with a ranking of very weak; it is possible that panelists were unable to pick up on the compound because it was coming off the column during the initial two-minute solvent delay by the GC–O panelists despite being detected by the GC–MS.

The ester profile of aging beers can undergo a drastic change due to continuous action associated by the yeast during bottle conditioning [34] and prolong storage [34,35]. As a beer ages, certain esters will decrease, while other ester will increase causing the beer to take on a more of a wine-like aroma [13]. One example is isoamyl acetate; it will start to hydrolyzed over time while the beer is being stored [22]. Although isoamyl acetate was identified in all the aged samples, it was not identified by any of the GC–O participants. Isoamyl acetate is commonly found in lambic beer but the levels that are found in lambics differ from that of traditional beers. In lambic beers, isoamyl acetate is found in much lower concentrations then in traditional beers. In traditional beers, isoamyl acetate can range anywhere from 1.2 to 2.8 ppm in lagers and 0.7 to 3.3 in ales [13,33]. SPME was used as the extraction technique throughout this experiment and has the capability of extracting compounds at the ppb level [36]. Although isoamyl acetate has a threshold level of 1.1–1.6 mg/L within beer, the concentration itself could have been below the threshold [37].

Medium-chain fatty acid ethyl esters, such as ethyl octanoate (caprylate), and ethyl decanoate (caprate), are traditionally found in lambic and gueuze beers. These ethyl esters are normally absent in lagers and present in only small concentrations in ales [26,38]. Both ethyl caprylate and ethyl caprate are considered to be typical aroma and flavor compounds of lambic and gueuze beer. Ethyl octanoate as well as ethyl decanoate were identified in all of the aging lambic beer samples using GC–MS but they were not identified by the panelists via GC–O. Individually, these compounds could be present below the threshold level; however, when combined, these compounds could both be perceived. This was

unexpected since ethyl hexanoate, ethyl octanoate, and ethyl decanoate give lambic and gueuze beer its wine and fruity flavor [13,39].

The secondary byproducts produced by the wild yeast strain *Brettanomyces* play a greater role in the aroma and flavor profile of lambic beer than traditional brewer's yeast, *Saccharomyces*. The main chemical compounds produced by *Brettanomyces* are the esters, ethyl acetate and ethyl lactate. Ethyl acetate and ethyl lactate can be formed enzymatically or chemically. Esterase is the enzyme used in the chemical reaction between ethanol and an organic acid to produce these esters. *Brettanomyces* displays a higher esterase activity than other yeasts (*Sacchromyces* or *Hanseniaspora*) [33].

*Brettanomyces* has been linked to the synthesis of ethylphenols (4-ethylphenol and 4-ethylguaiacol) and vinylphenols [40]. Brett is currently the only known microorganism linked to the development of ethylphenols. Some species of lactic acid have been known to produce ethylphenols in media but not in actual beverage systems [41]. *Brettanomyces* is a key player in the sensory profile of lambic and gueuze beer [2,42]. The "bretty" character is associated with a number of different aromas and flavors, such as mineral, tobacco, barnyard, leather, pharmaceutical and smoky (*20, 24*). Compounds produced by *Brettanomyces* have the ability to suppress a number of the desirable fruity ester aromas [43,44]. The horsey aroma can vary from slight to very strong. The strength of the horsey aroma is dependent upon the fermentation conditions; tetrahydropyridines are the compounds associated with the horsy smell. Tetrahydropyridines are produced from ethanol and the amino acid lysine from the wort [14,45]. Although 4-ethylphenol was found in all of the samples using GC–MS, it was not detectable by the panelists until the six-month sample. This is not surprising since *Brettanomyces*, although present in the beer early on, does not typically start producing detectable by-products (4-ethylphenol and 4-ethylguaiacol) until after the primary fermentation (*Saccharomyces)* has been completed [1].

Alcohols are produced as a byproduct of yeast metabolism. The concentrations of higher alcohols in lambic beers and other styles of beer are similar. Ethanol is just one of several alcohols that can be found in lambics along with higher alcohols (fusel alcohols), which are considered important by products of fermentation. Similar to ethanol and fusel alcohols, other alcohols are also produced during the main fermentation stage [21]. Based upon a study by Harrison, it was determined that a number of different alcohols can contribute to the flavor of beer: iso-amyl, phenethyl, propyl, and iso-amyl alcohol (2-methylbutanol) [46]. The higher alcohols identified do play an important role not only in the flavor of the beer, but also in the production of ester compounds as well.

Lambic and gueuze are well known for their high levels of acidity. It has been reported that gueuze has a lactic acid concentration of l500–3400 ppm and acetic acid had a concentration of approximately 700–1200 ppm [1]. Witrick et al. has found similar results based off of the commercial samples that where analyzed [7]. Ales and lagers have much lower acetic and lactic acid concentrations than gueuze beer. The concentration of acetic acid in ales and lagers tends to range anywhere from 60–140 ppm, while lactic acid in gueuze beer ranges between 70 and 200 ppm [38,46]. Lambic and gueuze beers are known for containing high levels of caprylic (C8) and capric (C10) acids. Capric acid concentration in gueuze beer usually exceeds 2 ppm, which is slightly higher than the concentration found in lagers or ales [26]. Isovaleric, hexanoic, octanoic, 2-ethylhexanoic, and decanoic acids were identified in the aging beer samples; however, only isovaleric, propanoic and hexanoic acids were detected by GC–O.

## 5. Conclusions

Lambic is a unique and complex beer. The results from this experiment show that lambics are composed of a number of volatile and semi-volatile compounds. However due to the extensive aging period and unique microbiota present in the beer, this can cause the chemical composition of the beer to gradually change over time. Chemical compounds that were once detected either by GC–MS or GC–O at one time point, may fall below detectable limits at the next sample period.

**Author Contributions:** K.W. and S.F.O. conceived and designed the experiments; K.W. performed the experiments; K.W. and S.F.O. analyzed the data; K.W. and S.F.O. wrote the paper. E.R.P. assisted by running the statistical analysis for this paper. All authors have read and agreed to the published version of the manuscript.

**Funding:** Funding for this work was provided in part by the Virginia Agricultural Experiment Station and the Hatch Program of the National Institute of Food and Agriculture, U.S. Department of agriculture.

**Conflicts of Interest:** The authors declare no conflict of interest.

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
