# Peer review of "Analysis of Lambic Beer Volatiles during Aging Using Gas Chromatography–Mass Spectrometry (GCMS) and Gas Chromatography–Olfactometry (GCO)"

_beverages, doi:10.3390/beverages6020031_

Round 1

Reviewer 1 Report

The subject presented in the manuscript is interesting and worthy of publication, but in my opinion, authors present a very preliminary research that should be significantly improved to be published.

Authors present a long section of individual results of volatile compounds identified by GCO for each ageing time. I think that all the presented results should be grouped into one single table, to facilitate their visualisation and comprehension. In stead of focusing in each of the ageing times separately, it would be more interesting to provide an overall description of the evolution of the aromas during ageing, and to detect significant differences of the compounds during this period.  

In addition, the methodology employed for GCO is not adequate. No quantitative information is given at all and therefore, it is difficult to perform an accurate treatment and discussion of the results, or any statistical treatment of them. In GCO there are many different methodologies employed to collect and process obtained data. One of the more simple one is called Frequency Detection methodology, and it easily provides with quantitative data that would allow a deeper discussion of the results. 

More over, a quantitative comparison of the results obtained by GCO and GCMS could also be performed. 

Finally, below are presented some other minor aspects to be improved:

- Introduction. Nothing about gueuze beer, ageing of beers, GCMS or GCO analysis of beers is mentioned. Please, include some additional information.

- Page 2, line 82. Please, indicate the temperature freezing.

- Page 3, line 99. Please correct “parametsrs”

- Page 3, lines 103-105. Please, indicate the standards employed for the training.

- Page 3, line 118. There is no mention of these conditions in the manuscript. Please, specify. 

- Table 2. Is there any difference between upper case or lower case X? Please, indicate it or correct the table.

- Lines 384-433. The format is different to the rest fo the manuscript. 

- Page 13. Remove Acknowledgments and appendix section if not employed. 

Author Response

Dear Reviewer 1,

I truly appreciate your comments and here is how I have tried to address them to the best of my ability.

  • The Acknowledges and appendix section was removed
  • All formatting issues were corrected
  • All tables were standardized to be uniform with upper case letters.
  • The GC-O conditions were added to the manuscript
  • Samples were stored in a -40°C freezer to ensure sample stability until analysis could be conducted.
  • In regards to panelists training the citation was added to the manuscript from the paper in which the training method was taken from.
  • An updated statement pertaining to the purpose of the project was added to the introduction along with the information about the gaining process of lambics.
  • Unnecessary tables where remove from the manuscript and combined into a single table. The data was reanalyzed using the detection of frequency method versus what was previously shown. After which the results and discussion sections were rewritten.

Reviewer 2 Report

Paper describes characteristic of volatile compounds analysed in lambic beer. Topic seems to be very interesting, although some chromatographical parts should be improved.

  1. I advice authors to perform some kind of quantification of compounds analyzed;
  2. lit. KI for ethanol is 427 (NIST17) whereas authors gave 587. Please clarify.
  3. Lit. KI for isopenthyl  is 736 (NIST17) whereas authors gave 700. Please clarify.
  4. Authors did not used chiral column. Please remove the stereochemistry mark in case of terpineol; 
  5. Which isomer of ethyl phenol authors meant? Or did they meant ethyl ether?
  6. Precise structure of Ethylhexanol;
  7. What about Amyl acetate, Nonanal  and heptanol in 28 M?
  8. Did authors have knowledge about  was the LOD or LOQ for the compounds?
  9. Sentence is not clear: " fatty-acid will split off from the alpha-acid molecule"
  10. Did the Ethyl acetate appears only in 9M? Please re-check that fact;
  11. As a supplementary data please add some sample chromatograms.

Author Response

Dear Reviewer 2,

I truly appreciate your comments and here is how I have addressed them.

  • I used a combination of calculated LRI values as well as several databases to confirmed LRI values. Hence why my LRI values are slightly different from that of the NIST library’s.
  • The stereochemistry was removed from the compound lists and just listed as alpha-ternineol.
  • In regards to the isomer of ethyl phenol was corrected to 4-ethylphenol
  • Ethylhexanol was corrected to 2-ethylhexanol
  • The GC-MS was corrected to be uniform and show the identification of the compounds in all of the months.
  • Ethyl acetate was identified in all of the GC-MS samples, however it was not detected by the panelists in all of the samples.

Round 2

Reviewer 1 Report

In my opinion, authors have improved significantly the paper and they have corrected all the requested issues. The methodology is now suitable and the results and discussion are deep enough to be published. 

However, they could add some references to the introduction section regarding analysis of volatile compounds in beers. This section could be extended in this way.

In addition, there are many mistakes in the numeration of the sections: Section 2.2. and section 3.1.1. are missing. Section 3.7 is not correct, GC parameters has no numeration, etc. Please, correct.

Author Response

I have made corrections based off of Reviewer 1's recommendation. I have fixed the numbering mistake found in the materials and methods as well as the results section. I have also expanded the introduction to include more information about the techniques used in the analysis of the volatile and semi-volatile compounds found in beer.

Reviewer 2 Report

I agree  with changes with authors.

Author Response

No changes were asked for by Reviewer 2